# Fairness Preferences in a Bilateral Trade Experiment

**Alice Ciccone [1,2,]*[ID], Ole Rogeberg [2,3] and Ragnhild Braaten [4]**

[1]    Institute of Transport Economics (TØI), Gaustadalléen 21, 0349 Oslo, Norway
[2]    CREE Frisch Centre, Gaustadalléen 21, Oslo 0349, Norway; ole.rogeberg@frisch.uio.no
[3]    Ragnar Frisch Centre for Economic Research, Gaustadalléen 21, Oslo 0349, Norway
[4]    Oslo Economics, Oslo 0349, Norway; rbr@osloeconomics.no
[*]    Correspondence: Alice Ciccone aci@toi.no; Tel.: +47-48263987

**Abstract:** Is the willingness to make trades influenced by how the total gains from trade are split between the trading partners? We present results from a bilateral trade game ($n = 128$) where all participants were price-takers and trading pairs faced one of three exogenously imposed trading prices. The fixed prices divided the gains either symmetrically in the reference treatment or asymmetrically in treatments favoring either the buyer or seller. Price treatments generating asymmetric gains from trade reduced desired transaction levels on both sides of the market, but more strongly by the disfavored party. The data weakly indicated a larger reduction when the disfavored party was a seller.

**Keywords:** fairness; inequality aversion; trade; markets; experiment; social preferences

**JEL Classification:** C90, D02, D03, D63

## 1. Introduction

Public concern with fairness in consumer and labor markets has been widely documented by surveys [1–3], boycotts or by increased demand for "fair trade" and charity-linked products [4–7]. These phenomena suggest that perceptions of fairness in terms of trade may influence the desire to trade. Some people may feel "exploited" by low wages or high prices; they may feel that they would exploit others if they bought products involving unfair trade with developing countries [8] (or, conversely, being willing to pay extra money for products made in ethical ways Elliott and Freeman [7]); or they may want to disassociate themselves from markets seen as legitimizing unfair actions. This latter concern has been for example used by opposers of emission quota trade systems [9].

In this article, we assess whether "unfair" imposed terms-of-trade in a quasi-market setting affects willingness to participate in trade that is Pareto-improving in monetary payoffs. While many experimental studies of other-regarding preference have involved institutions where strategic considerations play an important role [10], we reduce opportunities for strategic behavior to isolate the effect of other-regarding preferences[1]. Such preferences have previously been found in dictator games (where many individuals split the payoff with their co-player even in one-stage games, see e.g., Kahneman et al. [1] and Forsythe et al. [11].), but in these cases an increase in payoff to one player always comes at the expense of a decreased payoff for the other player. In our game, by contrast, both players maximize their own payoff as well as the other player's by trading to the maximum extent possible—but in two of the three treatments trading the maximum amount also maximize the inequality in the earnings distribution.

---

[1]    With other-regarding preferences we refer to individual preferences for another individual's payoffs, in addition to one's own.

Previous research has found that buyers in posted-offer experiments withhold Pareto-improving trade when posted prices implied an uneven distribution of the gains, but also that this effect decreased substantially in the final rounds. This suggests that the early restrictions were strategically motivated to force fairer offers in later rounds [12–14], indicating a smaller role for other-regarding preferences. Similarly, players are typically far more generous in the ultimatum game—where there are strategic motives—than in the dictator game—where there are none [11]. The importance of distinguishing between strategic and other-regarding preferences has also been highlighted in the literature on inequality aversion, where some theories highlight other-regarding preferences [15–17], while others focus on strategic motives [18–20].

We contribute to the literature by designing a stylised bilateral trade game that isolates other-regarding preferences. Strategic incentives or retaliation motives are largely absent: the terms of trade are fixed by the experimenters and no communication between partners in a trading pair is allowed. Our experimental design allows us to study how asymmetric splits of the gains from trade influence willingness to make trades that are Pareto improving in monetary payoffs, where the highest payoff for both participants is achieved by maximizing trade in each period.

Participants were randomly paired into *buyer-seller* pairs and randomized across three treatments defined by varying exogenously fixed trading prices. Sellers had high resource endowments in each period, but low productivity: they received a small payoff per resource held at the end of each period. Buyers had low resource endowments in each period, but high productivity: they received a high payoff from each resource held at the end of each period. The productivity difference generates a potential gain-from-trade, where the distribution of the gains depends on the transaction price per unit. Only the realized trade in each period, determined by the constraining partner(s) in the pair, was  made known to both participants.

Prices were exogenously given, both parties were price-takers, and all possible trades were Pareto improving in monetary outcomes. Both participants chose a desired transaction volume, and the lowest choice determined actual trade in the round. The game lasted ten rounds, and cumulative payoffs were updated between each round and displayed to participants both numerically and with bar-charts (see instructions in Supporting information Appendix B). In any round, participants were given the opportunity to test how these payoffs would change conditional on different trading volumes in the ongoing round.

Since all trades under all price regimes were Pareto-improving in monetary payoffs, both minimax, efficiency and strategic theories of inequality aversion would predict full trade, while the inequality aversion theory of Fehr and Schmidt [15] would predict trade restriction. This allows us to study whether "pure" inequality aversion influences willingness to trade in a trade setting where these potential confounds are removed by design.

Our hypothesis, in line with the theory of inequality aversion by Fehr and Schmidt [15], is that strongly asymmetric payoffs would reduce trade - particularly from the low-benefiting side. In an earlier pilot experiment conducted in Oslo, Norway, we also assessed whether low-gain sellers reduced trade more than low-gain buyers[2]. This hypothesis, for which we do not have a theoretical justification, seemed psychologically plausible to us based on introspection, in that having to sell something you own for a price far below its value would feel more unjust than having to buy something for a price close to its actual value.

The results show clear evidence of trade-restriction from both sides in treatments with asymmetric price, but a substantially stronger reduction from the low-benefit partner. The hypothesis that low-benefit sellers reduce more than a low-benefit buyer finds weaker support. Estimates are similar

---

[2]    The results from the pilot experiment are not used in the paper, as the design involved an unnecessarily involved sequential move mechanic that makes it difficult to compare the results.

when based on data from the last five rounds (to avoid potential learning effects), but weaker for the seller/buyer distinction.

The paper is organized as follows: Section 2 presents the game and gives a detailed description of the experimental design. Section 3 describes the data and reports the results. Finally we conclude by discussing the findings and their implications.

## 2. Experimental Design

A total of 128 students from Columbia University were recruited with the Online Recruitment System for Economic Experiments (ORSEE) and divided in six sessions over different days. The participant group is diverse in terms of gender, discipline of study and country of origin, although a majority of participants are from the United States (see Table 1). The experiment was programmed with z-tree software [21] and was run in April 2014 at Columbia Experimental Laboratory in the Social Sciences. The experiment used a fixed-partner and between-subjects design. Take home earnings ranged from USD 5.4 to USD 45.6 with an average of USD 24.3, plus a flat show up fee of 5 USD. The payment was made in cash immediately after the experiment as the students left the lab. Subjects were asked to answer a short survey immediately after playing the game. The experiment lasted in total about one hour. At the beginning of each experimental session, a written set of instructions was handed out to the participants and read aloud (reported in Supporting information Appendix B).

**Table 1.** Summary Statistics.

| Variable | Mean | Std. Dev | Percent |
|---|---|---|---|
| Female | | | 55% |
| Age | 24 | 3.9 | |
| Field of study: | | | |
|   Economics & Business | | | 15% |
|   Other social sciences | | | 30% |
|   Engineering | | | 11% |
|   Humanities | | | 9% |
|   Natural Sciences | | | 12% |
|   Other | | | 23% |
| Nationality: | | | |
|   North America | | | 48% |
|   South America | | | 5% |
|   Asia | | | 30% |
|   Europe | | | 7% |
|   Other | | | 11% |

Number of Observations: 128.

### 2.1. The Trade Game

Participants were randomly assigned to be *buyers* or *sellers* and then randomly matched into trading pairs that remained fixed for ten identical trading rounds. Subjects were told that they would face a trading price that was given and constant across all rounds, and that this price would determine how the gains of trade were split between the two parties. Players were also given a fixed number of resources as endowment every round, and a productivity parameter that translated resources into the Experimental Currency Unit (ECU). The ECU had a fixed and known exchange rate to US dollars, with 50 ECU = 1 USD.

Buyers converted resources to monetary payoffs with a high productivity, but were given a low initial resource endowment. In each round, they received one resource and at the end of each round they earned 20 ECU for each held resource. Sellers faced the opposite situation, with low productivity and high resource endowments: in each round, they received 20 resources and they earned 1 ECU per resource held at the end of the round. Note that the no-trade payoffs were identical (20 ECU per

round) for both buyers and sellers. The total gain from trading was determined by the difference in productivity, in that each resource held at the end of a round was worth 19 ECU more for the buyer than for the seller. Price, resource endowments and productivity for both types was common knowledge.

During the experiment, the buyer-seller pairs were free to trade resources at a fixed and given price to increase their payoffs. The instructions specifically reminded them that they could trade resources with their partner "to increase earnings". Trade in each round took place as follows: both players simultaneously stated their desired trading level (ranging from zero to ten units per round). Since trade was voluntary, actual trade was determined by the lowest offer. Before making their choice, both parties were visually informed, with bar graphs and numbers, of their own and their co-player's (1) secured incomes in the present round (i.e., the players' earnings in the absence of trade); (2) possible gains from different levels of trade in that round; and (3), from the second round onwards, cumulative payoffs from all previous trading rounds. For players in all treatment groups, this visualization made one aspect of trade particularly relevant to the experiment clear, that is, that trade increased both parties' payoff. The visualization received by players under the asymmetric treatment also made it clear that trade made outcomes highly unequal.

Prices were known, exogenously given, and constant over all rounds within each buyer-seller pair. Each pair faced one of three price treatments. The *buyer treatment* favors the buyer, with a fixed price of 3 ECU. For each unit sold, the seller made a 2 ECU profit relative to keeping the unit and producing ECUs with a low productivity, while the buyer made a 17 ECU profit by receiving 20 ECU from production minus the purchasing price of 3 ECU. The *seller treatment*, benefits the seller with a price of 18 ECU. Each unit traded in this treatment gave the seller a profit of 17 ECU (18 in sales price minus 1 in foregone production income), while the buyer netted a profit of 2 ECU. Finally, the *symmetric treatment* is equally beneficial to both with a price of 10.5 ECU. They both earned a profit of 9.5 for each resource traded (the seller earned the sales price 10.5 minus 1 in foregone production and the buyer earned the production "revenue" of 20 minus a purchasing price of 10.5 ECU). Payoffs are shown in Table 2. For instance, in the buyer treatment, if the buyer suggested full trade while the seller offered to trade only 8 resources, the actual number of resources traded would be 8 and the final payoffs would be 156 ECU for the buyer and 36 ECU for the seller.

**Table 2.** Payoffs for Buyers and Sellers by Treatments.

| Resources | Buyer T Price: 3 | | Seller T Price: 18 | | Symmetric T Price: 10.5 | |
|---|---|---|---|---|---|---|
| | Buyer | Seller | Buyer | Seller | Buyer | Seller |
| 0 | 20 | 20 | 20 | 20 | 20 | 20 |
| 1 | 37 | 22 | 22 | 37 | 29.5 | 29.5 |
| 2 | 54 | 24 | 24 | 54 | 39 | 39 |
| 3 | 71 | 26 | 26 | 71 | 48.5 | 48.5 |
| 4 | 88 | 28 | 28 | 88 | 58 | 58 |
| 5 | 105 | 30 | 30 | 105 | 67.5 | 67.5 |
| 6 | 122 | 32 | 32 | 122 | 77 | 77 |
| 7 | 139 | 34 | 34 | 139 | 86.5 | 86.5 |
| 8 | 156 | 36 | 36 | 156 | 96 | 96 |
| 9 | 173 | 38 | 38 | 173 | 105.5 | 105.5 |
| 10 | 190 | 40 | 40 | 190 | 115 | 115 |

## 3. Results

This section analyses trading behavior of 126 students from Columbia University who participated in six experimental sessions run at similar times during a two-week period in April 2014.[3] There was a

---

[3] The original sample was 128, but one pair is discarded because one subject reported no understanding of the game in the post-experiment survey.

total of three treatments: 22 pairs took part in the *buyer favoring treatment*, 20 pairs in the *seller favoring treatment*, and 21 pairs in the *symmetric treatment*.

### 3.1. Statistical Model

Each participant faced constant trading terms within the experiment, and had to decide how many of 100 potential transactions they wanted to accept on these terms. Assuming that the probability of rejecting any *single* transaction is constant at the individual level conditional on an individual's position in the market (buyer/seller) and treatment (exogenous price), the number of rejected transactions will be a binomial draw with an individual-specific rejection probability.

The individual-specific rejection probability may differ systematically with treatment, in addition to differing "non-systematically" due to heterogeneity in the baseline trading preferences of the participants. If the underlying heterogeneity in baseline trading preferences can be captured using the flexible beta-distribution, this implies that the number of rejected transactions will follow a beta-binomial distribution. Note that this imposes no strong structure on the preference distribution: It allows for a bi-modal distribution where people are either trading fully or not at all, as well as distributions where people are concentrated around a specific region of trade-rejection probabilities.

An intuitive illustration of the beta-binomial distribution is that we face a large number of jars that have different shares of red and white balls. The distribution of red-ball shares across the different jars is given by a beta-distribution. Each observation (i.e., total trades rejected by an individual across 10 rounds) consists of 100 balls drawn from a single randomly drawn jar.

The beta-distribution is parametrized in terms of an "average" probability and a dispersion parameter that captures the heterogeneity in baseline trading preferences. We assume that the treatment effects influence the average probability, and model the average probability using a logit formulation with additive coefficients capturing the systematic influences from market position and treatment:

- **Inequality**—Individuals in the symmetric payoff treatment are the reference category—all others have this parameter capturing the effect of being in an asymmetric payoff treatment. We hypothesize that asymmetric price treatments would reduce trade.
- **Relative loser**—In one of the two asymmetric payoff treatments the price benefits the seller, in the other the buyer. We expecte that the "relative loser" would restrict trade to a larger extent.
- **Seller side**—Based on an earlier pilot experiment in Oslo, Norway, we suspected that sellers might restrict trade to a larger extent than buyers when on the "losing side" of an asymmetric payoff treatment. We speculate that this might reflect a feeling that "your" resource is being "exploited" by someone else.

The model is estimated using Bayesian Maximum Likelihood (ML). To reduce overfitting to stochastic noise, the model is specified using a normal distribution for the effect priors. The effect priors are centered on 0 with a standard deviation of 0.5 standard deviation. This is a conservative prior that, roughly stated, asserts that effect sizes are unlikely to alter an individual's "baseline" probability of rejecting a transaction by more than $+/-$ 65%. By comparison, a "frequentist" (non-Bayesian) maximum likelihood model would be equivalent to Bayesian model with a prior assigning equal prior probability to any coefficient value from minus to plus infinity.

Analysing the data within this model has the benefit that we estimate all effects simultaneously using a statistical model that explicitly accounts for between-participant heterogeneity in trading preferences. In addition, the Bayesian priors help shrink effect estimates towards zero and reduce the risk of statistical overfitting to random noise. For robustness we analyze the data using non-parametric tests, and the results are in line with the ones reported below (see Supporting information Appendix C).

### 3.2. Estimation

The actual distribution of "total rejected transactions" within each treatment is shown in Figure 1, separately for each side of the market (see supporting materials for further plots of raw data).

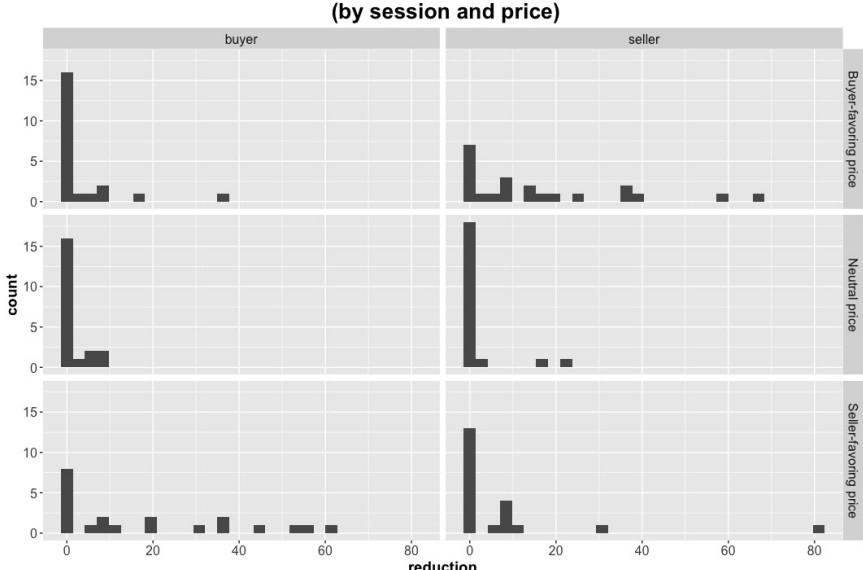

**Figure 1.** Each plot shows the observed distribution of "reductions from full trade" expressed by participants of a given type in a given treatment over all rounds combined.

The model is specified in the Stan language for probabilistic models (see supporting info for code in Supporting information Appendix A) and estimated using Hamiltonian Markov Chain Monte Carlo and a No U-Turn Sampler [22]. Four chains were run for 10,000 iterations, with the first half used for warm up and discarded from analysis.

*3.3. Results*

The estimation returns a sample of parameter values drawn from the posterior distribution of the model parameters. The model converges well, as shown by the Rhat values in the estimate table (Table 3). These assess how similar the posterior estimates are in different chains and in different sections of the same chain, and should be close to 1, as they are.

**Table 3.** Estimation output.

|  | Mean | SE_Mean | SD | 2.5% | 25% | 50% | 75% | 97.5% | n_eff | Rhat |
|---|---|---|---|---|---|---|---|---|---|---|
| $\phi_{raw}$ | −3.28 | 0.00 | 0.30 | −3.88 | −3.48 | −3.27 | −3.07 | −2.70 | 12,773 | 1 |
| $\lambda_{raw}$ | 0.66 | 0.00 | 0.21 | 0.23 | 0.52 | 0.67 | 0.81 | 1.06 | 15,810 | 1 |
| $seller_{coef}$ | 0.26 | 0.00 | 0.31 | −0.36 | 0.05 | 0.26 | 0.47 | 0.87 | 16,425 | 1 |
| $losing\_side_{coef}$ | 0.75 | 0.00 | 0.30 | 0.16 | 0.55 | 0.75 | 0.95 | 1.32 | 15,518 | 1 |
| $inequality_{coef}$ | 0.56 | 0.00 | 0.30 | −0.03 | 0.36 | 0.56 | 0.77 | 1.17 | 13,942 | 1 |
| lp | −2262.79 | 0.02 | 1.60 | −2266.80 | −2263.60 | −2262.47 | −2261.61 | −2260.69 | 8841 | 1 |

To assess the results, we compare the prior distribution imposed on the effect coefficients to the posterior distribution. This tells us how informative the data is - the extent to which it updates and shifts the prior. As seen in Figure 2, all effect distributions are updated in the expected direction. Given the model and the prior, the probability that the effect coefficients are positive has increased from 50% to 99.3% (the "relative loser" parameter), 96.9% (the "asymmetric price" parameter) and 80.1% (the "seller" coefficient)—with average effect parameter values across the posterior of 0.75, 0.56 and 0.26 respectively.

To interpret the parameters, we note that these influence the average transaction rejection probability through an additive term in a logit specification. Exponentiated parameters will express the odds-ratios of rejecting transactions when the effect represented by a given parameter is present

relative to when it is not. These odds-ratios can in turn be interpreted as relative risks, since the baseline rejection probability is very low.

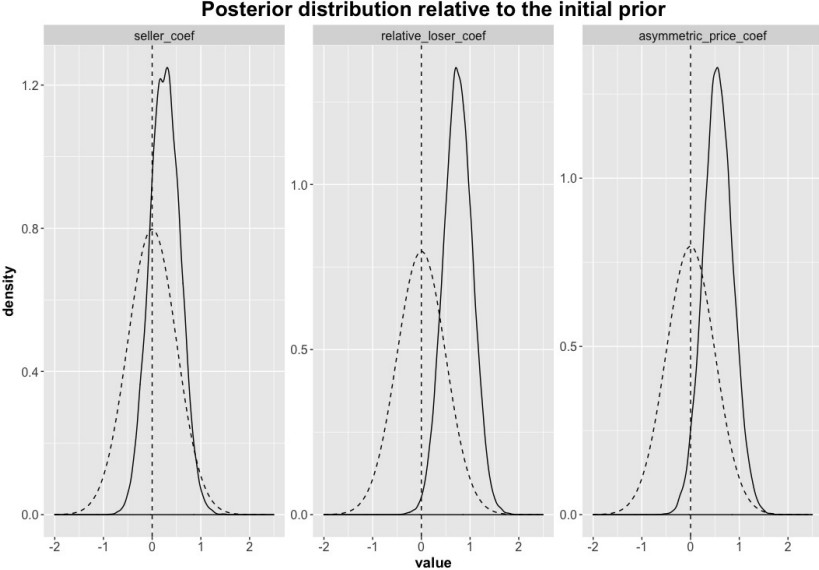

**Figure 2.** The distributions shown with dashed lines are the imposed effect prior, centered around 0. The solid line distributions show the posterior distribution, which is the updated effect distribution that conditions on the observed data.

We can thus calculate the relative risk of rejecting a transaction for the three possible combinations of effect parameters. Using the average parameter values under the posterior:

- Being in an asymmetric treatment on the benefiting side increases your probability of rejecting a transaction by a factor of 1.75
- A buyer with an asymmetric price that does not favor him/her has a rejection probability increased by a factor of $1.75 \times 2.12 = 3.71$
- A seller with an asymmetric price that does not favor him/her has a rejection probability increased by a factor of $1.75 \times 2.12 \times 1.30 = 4.823$

Note that these are relative risks, which means that the increase in absolute risk will be small for most participants given the low baseline trade-rejection probabilities (see Figure 3).

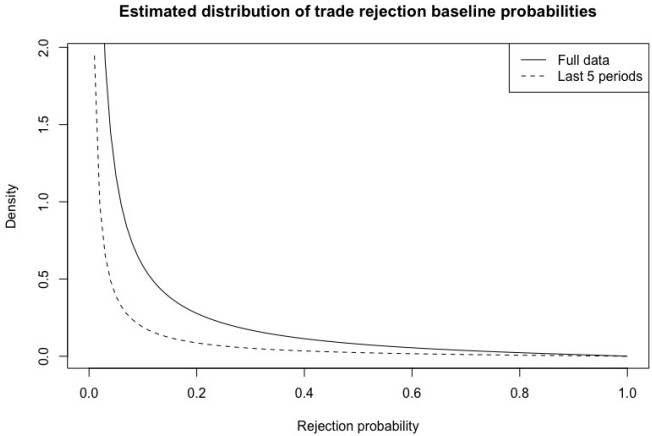

**Figure 3.** The plots show the estimated distribution of trade rejection probabilities using the mean values of the estimated beta coefficient from the posterior distribution. Estimates using the full data set and the last five rounds of the experiment are shown separately.

The model assumes that the individual-level rejection probabilities of the participants are fixed throughout the experiment. This may not hold. In particular, we did not have practice rounds and may suspect that data from the initial rounds reflects learning and "noise". Such noise would be common to all treatments and should not substantively influence the effect estimates, but would tend to increase the estimated dispersion of the baseline trade preferences.

To assess this, we re-estimate the model using only data from the last five rounds of the experiment. The effect coefficients remain similar (see Figure 4). We can also visualize the distribution of the baseline "non-trading preferences" in our sample. These are given by the parameters $\phi_{raw}$ and $\lambda_{raw}$ in Table 3, and are the logged values of the expectation and dispersion parameters from our beta distribution for non-trade preference heterogeneity [4]. In the main analysis, for instance, the median value for this parameter across the posterior implies a probability of not trading a single unit equal to $\exp(-3.48) =$ 3.1%. Plotting the preference distribution estimated from the full dataset and from the last five periods shows that the estimated heterogeneity of the baseline trading preferences is substantially reduced in the last five periods (Figure 3). This suggests that some of the non-trades chosen in the earlier periods can be viewed as learning, experimentation or "noise".

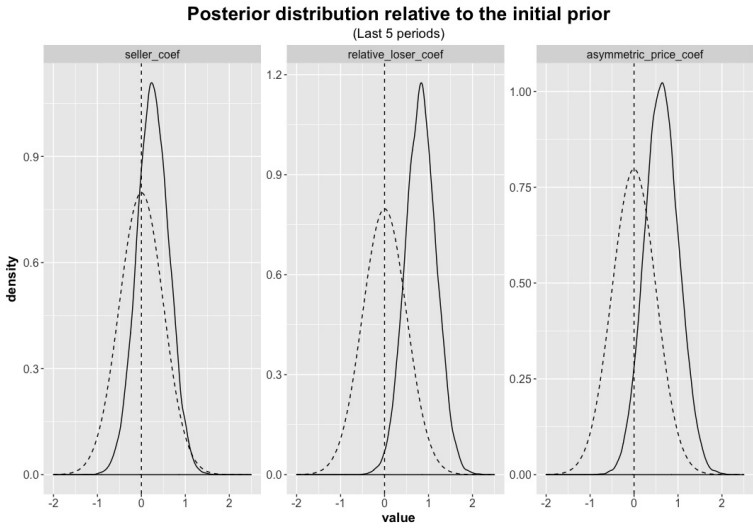

**Figure 4.** The distributions shown with dashed lines are the imposed effect prior, centered around 0. The solid line distributions show the posterior distribution, which is the updated effect distribution that conditions on the observed data from the last 5 rounds of the experiment.

Finally, we teste the inference procedure by re-estimating the model on scrambled data, created by randomly assigning treatment group labels to the observations. In this data there will be no systematic difference between the observations grouped under each treatment. In line with this, the posterior distributions for the effect parameters tends to remain centered on zero, with a reduced variance relative to the prior (not shown). Essentially, in data with no treatment effect the model correctly infers that we should be more certain that the treatment effects were close to zero.

## 4. Conclusions

In line with predictions from the Fehr and Schmidt model [15] we report evidence of substantial reductions in desired trade when the terms of trade generate substantial inequality. At the individual level, however, trade reductions were typically partial, while the linear utility formulation of the Fehr and Schmidt model would predict that any single participant would trade fully or not at all.

---

[4]　The most common way of parameterizing the beta distribution involves an $\alpha$ and a $\beta$ parameter. In terms of our parameters, $\alpha = \phi \times \lambda$ and $\beta = (1 - \phi) * \lambda$

While both parties to the trade reject more transactions when trade generates substantial inequality, the reduction is roughly twice as large for the party that benefits the least. While the prior shifts in the expected direction also for the "seller coefficient", indicating that a non-benefiting seller reduces transactions more than a non-benefiting buyer, the evidence of such role-asymmetric effects is weaker.

We note that all trades are Pareto improving in monetary outcomes, such that models of inequality aversion focusing on maximin payoffs or efficiency [16,17,23] would have predicted full trade. We also note that there are no obvious strategic motives for reducing trade in the experiment. In fact, the only way to increase the payoffs of both yourself and your co-player is to increase trade volume, and trading terms are fixed across all rounds for any participant. Only a minimal "signal", concerning the desired trade relative to the co-player, passed between players during the game. The lack of direct strategic motives in the experimental design supports an interpretation of the findings as aversion to an unequal distribution of trade outcomes.

The strong effect of "unfair" trading terms on the willingness to trade goes against the rather strong claim in Charness and Rabin [16] that "few subjects sacrifice money to reduce inequality by lowering another subjects' payoff, and only a minority do so even when this is free". The results imply that such factors *may* have an influence in some real world markets. We would not, however, claim that the current results are sufficient to *establish* a substantial role for such influences in real world markets. The experiment made the distribution of gains-from-trade highly salient to participants, whereas such information may often be missing in real world markets. Fair trade products, however, may be viewed as providing such information about the gain distribution in order to attract consumers who care about outcome inequality, arguably raising consumer welfare by facilitating product differentiation [4]. Also, we note that our experiment imposes an exogenous trading price, whereas real world markets have endogenous prices. In theory, this will allow trading parties to agree on an acceptable distribution of the gains from trade and realize all Pareto-improving trades. In practice, however, workers in large companies and consumers in grocery stores may be unlikely to believe that they can enter into bilateral negotiations over wages and prices. As a result, markets may be perceived by participants at different ends of the value-chain as having "exogenously fixed" terms of trade, and the decision to participate in the market or not may then be influenced by perceived "fairness" of the current terms-of-trade.

**Author Contributions:** conceptualization, O.R. R.B. A.C.; methodology, A.C. O.R. R.B.; formal analysis, A.C. O.R.; investigation, A.C.; resources, R.B. A.C. O.R.; data curation, A.C., O.R.; software, A.C; writing– original draft preparation, A.C. O.R. R.B.; writing– review and editing, R.B. A.C. O.R.; project administration, R.B. A.C. O.R.; funding acquisition, R.B. O.R.; please turn to the CRediT taxonomy for the term explanation. All authors have read and agreed to the published version of the manuscript.

**Funding:** This research was funded by Oslo Centre for Research on Environmentally friendly (CREE), the Norwegian Research Council, the Department of Economics (UiO) and the project "International cooperation on emission abatement" (Norwegian Research Council, project number 193703).

**Acknowledgments:** We wish to thank all those who helped us with feedbacks and valuable comments: Ernesto Reuben, Kjell Arne Brekke, Fortuna Casoria, Stephan Kroll, Peter Martinsson, Alberto Antonioni, Timo Goeschl; and participants in workshops organized by the mentioned NRC project.

**Conflicts of Interest:** The authors declare no conflict of interest. The funders had no role in the design of the study; in the collection, analyses, or interpretation of data; in the writing of the manuscript, or in the decision to publish the results.

## Appendix A. Supporting Information: Model Code

```
data {
  int<lower=0> obs_count;
  int<lower = 1> transaction_count;
  int proposed_trade_reductions[obs_count];
  int inequality[obs_count];
  int buyer[obs_count]; // 1 if a buyer, 0 otherwise
  int losing_side[obs_count];
}
parameters {
  real phi_raw; // Distributional parameters for the initial trading pref
  real lambda_raw; // Distributional parameters for the initial trading pref

  real seller_coef;
  real relative_loser_coef;
  real asymmetric_price_coef;

}
model {
  real temp_term;

  // priors
  phi_raw   normal(0,3);
  lambda_raw   normal(0,2);
  seller_coef   normal(0, 0.5);
  relative_loser_coef   normal(0, 0.5);
  asymmetric_price_coef   normal(0, 0.5);

  for (i in 1:obs_count){
    temp_term =  phi_raw + ((inequality[i] == 1)? asymmetric_price_coef : 0)  +
            ((losing_side[i] == 1) ? relative_loser_coef : 0) +
            ((buyer[i] != 1 && inequality[i] == 1 && losing_side[i] == 1)? seller_coef : 0);

    proposed_trade_reductions[i]   beta_binomial(100, inv_logit(temp_term) * exp(lambda_raw),
    (1-inv_logit(temp_term)) *exp(lambda_raw));
  }
}
```

## Appendix B. Supporting Information: Instructions

# Instructions

## General information

Welcome to this economics experiment. You will now make anonymous decisions on a computer – neither the researchers nor the other participants will ever know which decisions were made by which participant. Communication with other participants during the experiment is not allowed. If you have questions or need help with the computer, please raise your hand and one of us will approach you and assist you privately.

It is important to us that you trust the information we give in this experiment, so we want to emphasize that *all the information you receive is true*. It is also important that you fully understand how the experiment works, so we ask you to read the instructions carefully.

## Instructions

During the experiment you will be asked to make decisions regarding trade, in terms of ECU points (Experimental Currency Unit) that will be exchanged into US dollars and paid out at the end of the experiment.

<div align="center">Exchange rate: 50 ECU= $1</div>

Before the experiment starts you will be randomly placed into pairs. Within each pair, one individual will be randomly assigned to be a *seller* while the other one will be a *buyer*. Nobody can learn the identity of their co-players, and no communication with the co-player is possible. The pairs remain fixed throughout the experiment.

The experiment consists of 10 identical periods. At the beginning of each period you will receive a fixed sum of resources. Buyers and sellers differ in the number of resources they receive and in their productivity, which determines how much money they can earn from production using these resources.

During the experiment, resources can be traded with your co-player to increase earnings (details regarding the trade follow). All trade will occur at set price determined at the beginning of the experiment. This price will be the same in all rounds and it will be presented on the first screen.

At the end of each period you will receive a certain number of ECU based on the number of resources you are currently holding, your productivity and the payments or earnings due to trade. Buyers earn 20 ECU for each resource unit held after trading while sellers earn 1 ECU for each resource held after trading. In other words, a buyer produces 20 times as many ECU per resource unit then a seller.

The following table summarizes the relevant information.

|  | **Buyer** | **Seller** |
| --- | --- | --- |
| Number of resources received every period | 1 | 20 |
| Productivity (Number of ECU earned for each resource held at the end of a period) | 20 | 1 |

Briefly told, you will be given a certain amount of resources at the beginning of each period. You and your co-player will then choose how many resources to trade. When trading is over, each player's earnings for that period will be determined by the amount of resources they hold, their productivity, and the payments made or earned from the traded resources. For instance, imagine that a buyer can buy resources at a price of 10 ECU. Each resource would then produce 20 ECU to the buyer while costing 10 ECU. *The net payoff for the buyer would then be 10 ECU per unit bought.* In the same trade, the co-player, the seller, would earn 10 ECU per resource sold, while losing 1 ECU in foregone production. *The net gain for the seller in this period would be thus 9 ECU per resource.*

**Trade is organized as follows:**

Both players in a pair state how many resources they wish to trade between 0 and 10. Since trade is voluntary, the lowest of the two numbers will determine the actual amount traded. For instance: if the buyer wants to buy 5 units while the seller wants to sell 6 units, then the seller will only be able to sell the 5 units that the co-player wants to buy.

To help you decide, you will be able to visualize the outcome resulting from possible trades by clicking on the "visualize" button. This will display two bars representing your total payoff in the case the proposed trade is going to be the actual trade. The bars highlight your accumulated payoff from previous periods, the payoff resulting from the number of resources hold in a period and the payoff resulting from trade in that period. An example is illustrated below.

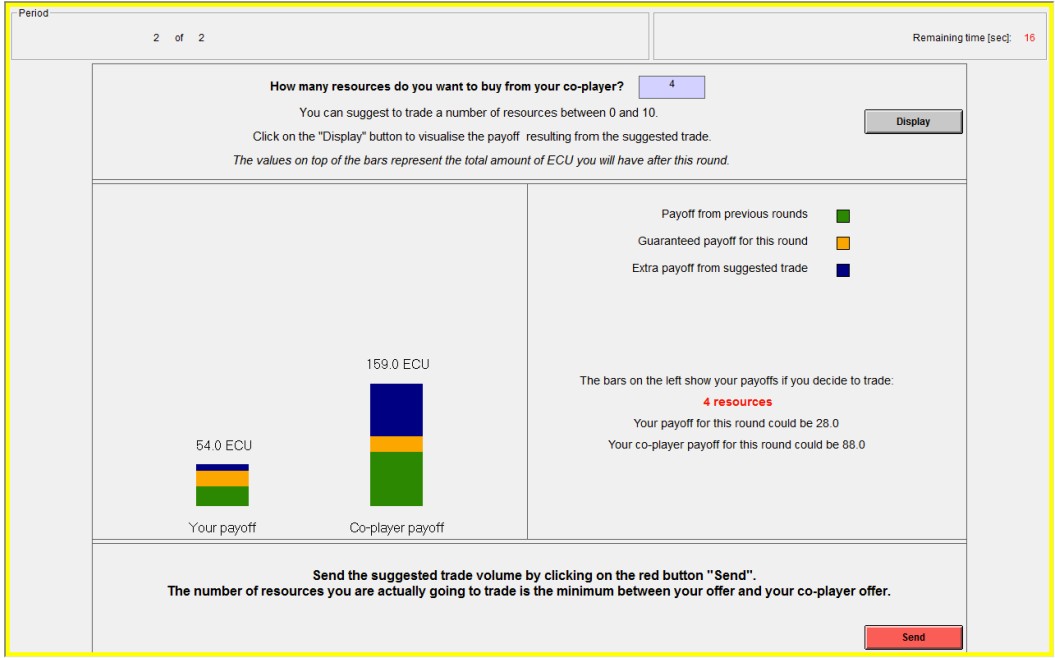

You have to visualize the outcome of at least one proposed trading volume before you can send your actual decision. Your co-player will not know which trading level (or how many) you examine before making your actual choice. When you have decided how much you wish to trade, click the "send" button. In a screen like the one below, you will then learn how many resources were traded and how the trade has affected the earnings you and your co-player have made so far in the experiment.

It is important to note that the *axes on the columns changes from round to round,* since they also display how much you have already earned. To improve clarity, above each column you will also see the **total** number of ECU displayed by the column.

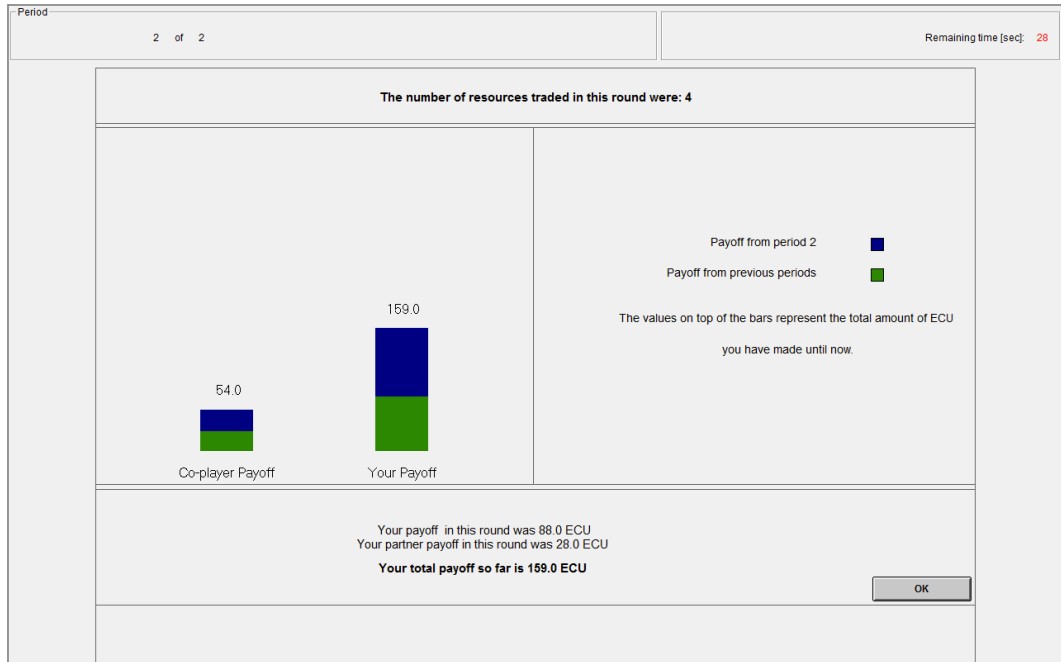

### Periods

This is how one period of the experiment is run. There will be 10 such periods.

When the last round is over, we will ask you to answer a survey. Note that this survey, like the experiment, is anonymous.

### Questions?
If you have any questions please raise your hand and we will assist you privately.

### Appendix C. Supporting Information: Hypothesis Tests

We now report the results obtained using standard non-parametric analysis testing. These insights show the robustness of our results to different methodological approaches.

*Appendix C.1. Full Trade in the Symmetric Treatment*

In the symmetric treatment, the price was set such that gains from trade were split equally between buyer and seller within each trading pair. When payoffs were symmetric and inequality between parties is absent, subjects quickly maximized their payoffs by trading the maximum number of resources possible (i.e., ten resources).

Figure A1 shows how, in the symmetric treatment, an upwards trend reaches full trade after a few rounds. The survey run at the end of the experiment suggests that some students used few rounds at the beginning of the experiment to understand the game and to test different trading volumes.

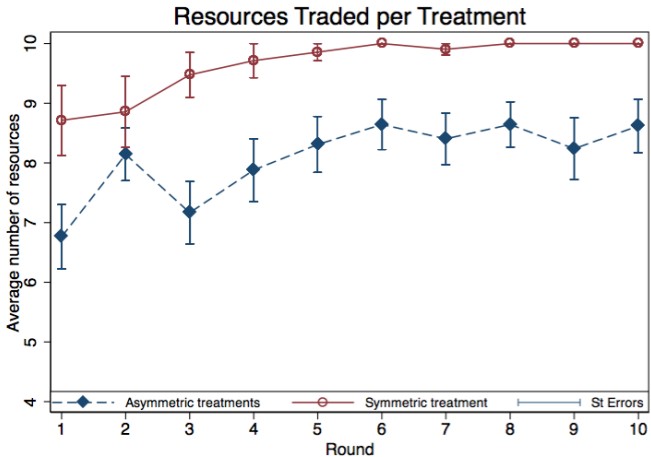

**Figure A1.** Mean actual number of resources traded per round in the symmetric and asymmetric price treatments, with standard errors. The buyer and seller treatments are merged into one asymmetric treatment. The figure shows that trade in the symmetric treatment quickly converges to full trade, while the number of resources traded in the asymmetric treatments is inferior.

We run non-parametric tests considering all ten rounds as well as discarding the first four, five, and six rounds to discard learning effects. The average number of resources traded in the symmetric treatment was 9.65 when considering all ten rounds and 9.93, 9.96 and 9.98 when considering the last seven, six, and five rounds respectively (Table A1). Only one third of the 21 pairs in the symmetric treatment ever deviated from full trade. Of these seven buyer-seller pairs, only one failed to reach full trade within the fourth round. In other words, only 1 out of 21 participant pairs traded less than the full amount after the fourth round, and even this pair chose full trade in the last three rounds.

*Appendix C.2. Trade is Significantly Restricted in the Asymmetric Treatment Compared to the Symmetric Treatment*

There is significantly less trade in the asymmetric treatment compared to the symmetric treatment (Figure A1). When pooling data from the asymmetric treatments and considering all rounds, the mean trading volume is 8.08 in the asymmetric treatments and 9.65 in the symmetric treatment (Table A1). These trade restrictions are the result of 32 pairs (76%) in the asymmetric treatment restricting trade at least once in the ten rounds played, while only seven pairs (33%) have trade reductions in the symmetric treatment. If we consider only the last six rounds of the game to eliminate possible learning effects, the number of pairs trading less than ten resources in the asymmetric treatment decreases to 18 (43%) bringing the average trade in the asymmetric treatment down to 8.51. In the symmetric

treatment the average trade is 9.98 because only one pair (4.7%) restricted trade in the last six rounds of the experiment (Table A1).

**Table A1.** Average Actual Trade in Asymmetric and Symmetric Treatments.

| Considering | Asymmetric Ts (N = 42) | Symmetric T (N = 21) | Wilcoxon P-Val |
| --- | --- | --- | --- |
| 1st–10th round | 8.08 | 9.65 | **(0.0002)** |
| 4th–10th round | 8.39 | 9.93 | **(0.0002)** |
| 5th–10th round | 8.47 | 9.96 | **(0.0006)** |
| 6th–10th round | 8.51 | 9.98 | **(0.0015)** |

Non-parametric two-samples Wilcoxon rank-sum (Mann-Whitney) tests are used to test the null hypothesis of equality between the distributions of the asymmetric treatment and the symmetric treatment. All reported p-values imply the rejection of the null hypothesis. the number of observations refers to "pairs", hence 42 pairs when considering both asymmetric treatments and 21 pairs in the symmetric treatment.

Non-parametric two-sample Wilcoxon rank-sum tests reported in Table A1 suggests that the null hypothesis, equality of the means between the asymmetric treatment and the symmetric treatment, can be rejected with a high statistical significance independent of the number of rounds considered (p-value between 0.0015 and 0.0002). Moreover, the reduction observed in actual trading volumes in the asymmetric treatment creates an efficiency loss of about 15% (16.3% considering all rounds and 14.7% considering the last six rounds) compared to trade occurring in the symmetric treatment.

*Appendix C.3. Result 3: The Worse-Off Party Within a Pair Is the Main Driver of the Observed Trade Restrictions in the Asymmetric Treatment*

The disfavored party within a trading pair is found to be the main constraint on trading volumes.[5] The average number of resources suggested for trade by the worse-off party is between 12% and 9% lower than the average trading proposals of the better-off party in all rounds (Figure A2).

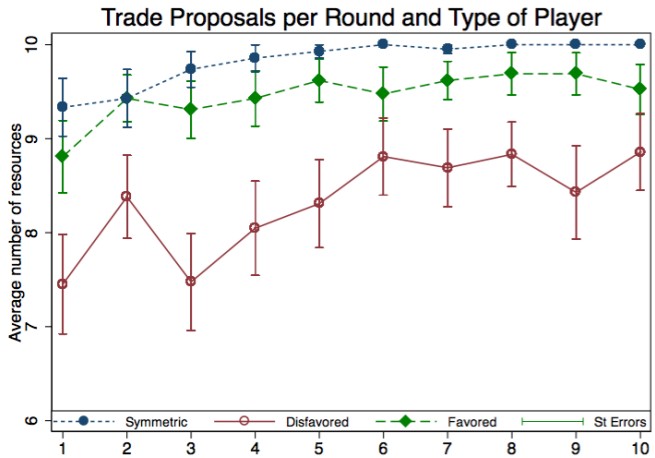

**Figure A2.** Mean trade proposals per round and type of player in the symmetric and asymmetric price treatments with standard errors. The disfavored party is defined as the buyer in the seller-favoring-treatment and the seller in the buyer-favoring-treatment and vice-versa for disfavored party.

Table A2 reports the individual suggested mean trade when all ten rounds are considered and when only the last seven, six, and five rounds respectively are considered, together with the p-values from the non-parametric test in parenthesis.

---

[5] We define buyers and sellers in the symmetric treatment as "symmetric" players.

**Table A2.** Average Trade Proposals by Type of Player.

| Obs | Favored Pl. (N = 42) | Disfavored Pl. (N = 42) | Symmetric Pl. (N = 21) |
|---|---|---|---|
| 1st–10th round | 9.46 | 8.33 | 9.82 |
| Wilcoxon p value | (0.5372) | **(0.0009)** | |
| 4th–10th round | 9.58 | 8.57 | 9.96 |
| Wilcoxon p value | (0.3304) | **(0.0038)** | |
| 5th–10th round | 9.60 | 8.65 | 9.98 |
| Wilcoxon p value | (0.3304) | **(0.001)** | |
| 6th–10th round | 9.60 | 8.72 | 9.99 |
| Wilcoxon p value | (0.1816) | **(0.0004)** | |

The non-parametric Wilcoxon rank-sum tests the null hypothesis of equality of means respectively between the trade proposals of the favored and disfavored players in the asymmetric treatment, and the symmetric players in the symmetric treatment. P-values in bold imply the rejection of the null hypothesis.

Average offers of disfavored players in the asymmetric treatment are significantly lower than average offers of players in the symmetric treatment, no matter the number of rounds considered (two-sample Wilcoxon non-parametric tests, p-values between 0.0038 and 0.0004). On average, favored players suggest lower or equal trade in every round, but these differences are not statistically significant (Table A2). In the non-parametric tests reported in Table A2, the unit of observation for the asymmetric treatment is the average suggested trade for each type of player (either favored or disfavored), while in the symmetric treatment is the average suggested trade of each pair.

*Appendix C.4. Result 4: A small share of favored subjects also restrict trade.*

If favored players are unaffected by the price treatments, their probability of proposing less than full trade should be the same as in the symmetric treatment. There are some indications that the favored party in the asymmetric treatments sometimes reduce trade (Figure A2).

The experimental data show that a small number of favored players also suggest to trade less than ten resources and this number is higher than in the symmetric treatment. To test the independence of the observed distribution we use the Chi-squared test with one degree of freedom. The results are significant at the 10% level, but they are sensitive to the number of rounds considered (Table A3). Given the small number of observations we also ran Fisher's exact test, which delivered similar results (and it is hence not reported here).

**Table A3.** Trade Restrictions by Type of Player and Treatment.

| | Favored Pl. (N = 42) | Symmetric Pl (N = 42) | Pearson Chi2 DF = 1 |
|---|---|---|---|
| 1st–10th round | 16 (38%) | 9 (21%) | 2.79 **(0.095)** |
| 4th–10th round | 7 (17%) | 1 (2%) | 4.97 **(0.026)** |
| 5th–10th round | 5 (12%) | 1 (2%) | 2.87 **(0.090)** |
| 6th–10th round | 5 (12%) | 1 (2%) | 2.87 **(0.090)** |

Number of individuals who suggested trading less than ten resources at least once. We consider all ten rounds and the last seven, six, and five rounds. The last column reports Person Chi-squared with one degree of freedom and p-values in parenthesis.

It is worth noticing that among the 16 better-off players who suggested at least once to trade less than ten resources in all ten rounds, twelve actually restricted trade, i.e., they were the party who suggested the smaller number of resources in at least one round. If we look at the last five rounds only, four favored players, out of five who suggested less than ten resources, actually constrained trade. In the symmetric treatment, eight out of nine restricted trade in the first few rounds and only one player actively restricted trade after the fourth round.

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
