# Peer review of "Fairness Preferences in a Bilateral Trade Experiment"

_games, doi:10.3390/g11010008_

Round 1

Reviewer 1 Report

The paper is clearly and efficiently presented and organized. The experiments are clearly presented, the results are clear. The appendices to the paper are standard in this type of literature. I have no major comments.

I would suggest to change the title, which I find misleading: the paper is really on unfairness in exchange as a limit to trade, not really on the trade-off between inequality and efficiency.

Author Response

Thank you for taking your time to review our paper and for your positive feedback.

We agree with your suggestion to change the title and have changed it line with your suggestion.

The new title we can suggest is “Fairness preferences in a bilateral trade experiment”, which was the title of an earlier version of the paper published as a working-paper.

We hope you agree that this title better fits the focus of the paper.

Best regards

Reviewer 2 Report

A very nice paper.  I only have a few editorial comments

I suggest using the present verb tense in all parts of the paper except when describing the experiment.

In section 2, a table of summary stats would help illustrate the diversity of your sample.

line 22:  You may want to clarify what "other-regarding preference" 

line 20: you may want to start "In this paper, we assess ..." so the reader knows you are talking about your study.

line 46: replace different with varying

line 126-127: The seller treatment benefits the seller with a price 18 ECU.

line 129: The symetric treatment is equally beneficial to both with a price of 10.5 ECU.

You use the term 'pair' on page 3 and then switch to 'couple' on page 4.  I would use 'pair' throughout the paper.

Author Response

Thank you for the feedbacks.

We have changed the text according to your suggestions, as explained as follow:

I suggest using the present verb tense in all parts of the paper except when describing the experiment.

We have changed it throughout the paper except when we refer to specifications of the experiment, as suggested.

In section 2, a table of summary stats would help illustrate the diversity of your sample.

This is now Table 1. We have included: gender, age, field of study and nationality.

line 22: You may want to clarify what "other-regarding preference"

See footnote 1.

line 20: you may want to start "In this paper, we assess ..." so the reader knows you are talking about your study.

done

line 46: replace different with varying

done

line 126-127: The seller treatment benefits the seller with a price 18 ECU.

done

line 129: The symetric treatment is equally beneficial to both with a price of 10.5 ECU.

done

You use the term 'pair' on page 3 and then switch to 'couple' on page 4. I would use 'pair' throughout the paper.

done

Best regards